# OpenReview forum: "Beyond Binary Evaluation: Measuring Language Model Hallucinations Through Distributional Correctness"
_ICLR.cc/2026/Conference — Submitted to ICLR 2026_

### Official Review · Reviewer_Wvmz · 2025-10-27

**Soundness:** 2
**Presentation:** 3
**Contribution:** 2
**Rating:** 2
**Confidence:** 4

**Summary:**

Under binary scoring (+1 for correct, 0 for incorrect), there is never a reason to abstain from answering or say "I don't know": it's always better to make a guess. It is then surprising that LLMs trained to perform well under binary scoring learn to "hallucinate", i.e., make a best guess even when they know. The paper proposes a new metric called the Distributed Correctness Score (DCS) that rewards abstention when the model is uncertain. The authors discuss similarities and differences between DCS and existing uncertainty-aware metrics like confidence-weighted accuracy. The authors evaluate 6 LLMs on 12 benchmarks and show that even modern LLMs perform quite poorly with respect to DCS: for 6 of the benchmarks, all LLMs have negative scores.

**Strengths:**

I think the issue of binary scoring incentivizing guessing is crucial, not just for hallucination but for all forms of caution and overconfidence in LLMs. Incorporating a correctness metric which penalizes overconfidence is a natural countermeasure. The selection of models and benchmarks to test is reasonable. Scores are reported in a clear way, with standard error included. The writing is quite clear throughout.

**Weaknesses:**

### **Concerns about the DCS metric**

I have major concerns about the DCS metric which the whole paper revolves around. I agree that this metric solves the core issue with binary scoring, but it seems to me that proper scoring rules solve this issue better. For the area chair, DCS is defined as $(\ell_c p_c - \ell_W p_W) (1- p_{IDK})$ where $p_c,p_W, p_{IDK}$ are the probabilities the model respectively assigns to the correct answer, to all of the wrong answers in aggregate, and to the "I don't know" answer. $\ell_c$ and $\ell_W$ are user-chosen parameters that are usually treated as 1.

1. It seems to me that a rational agent will always choose either p_{IDK} = 0 or p_{IDK} = 1. Specifically, it is optimal to choose p_{IDK} = 0 iff the model believes that $(\ell_c p_c - \ell_W p_W) > 0$. As such, DCS does not elicit the true belief state of a rational agent.

2. Why is it natural to include an explicit IDK option, rather than eliciting a distribution over the actually plausible answers and then deciding whether to abstain based on that distribution? The model knows that IDK is not the correct answer, so the model's true belief state over the correct answer should answer 0 probability to IDK.

3. The authors critique proper scoring rules by saying that they "fail to capture the full richness of a model’s belief state": specifically, proper scoring rules only consider the max probability of any answer and ignore how the rest of the probability mass is distributed. However, I am unconvinced that the rest of the distribution matters. The authors' argument seems to be that we care how much probability is assigned to IDK. But this issue goes away if IDK is not included as an answer option to begin with.

4. The authors also argue:
> Unlike forecasting tasks where the ground truth is a stochastic label, language model evaluations present deterministic facts of the matter. The ‘true’ conditional distribution is a point mass, so it is meaningless to demand that a model report the frequency of correctness for each option, as for example the Brier score rewards...Our objective is not to elicit calibrated probabilities but to measure trustworthy epistemic behaviour.

While the authors mention one interpretation of the Brier score, it is perfectly well-defined to still award of a score of 1 - (p-c)^2 where p is the max probability of any answer and c in {0,1} indicates whether that answer is correct. Furthermore, this definition maintains the desirable property that a rational agent should always report its true probability distribution over answers. DCS does not have this property, as mentioned in Point 1 above. It's also unclear why calibrated probabilities are not satisfactory as "trustworthy epistemic behaviour", or why DCS induces more trustworthy epistemic behaviour.

**What I would find convincing.** The main argument I see for DCS over proper scoring rules is that if we want to finetune models to sometimes abstain, then it makes sense to include an explicit IDK option in the finetuning dataset. This argument makes sense to me, but no finetuning experiments are performed to see whether DCS effectively serves this purpose. Also, a simpler scoring rule like "+1 for correct, 0 for abstain, -1 for wrong" could suffice for the finetuning application. See [Kang et al (2025)](https://aclanthology.org/2025.naacl-long.183/), which performs RL finetuning on this simpler scoring rule and shows that it successfully teaches the model to abstain. I think it's totally plausible that finetuning on DCS is more effective at teaching models to appropriately abstain. But if that's the main claimed benefit of DCS, then I think the paper needs to show that experimentally.

## **Other issues**

1. No proper scoring rules are included in the experiments as baselines.
2. The authors mentioned that in two cases, a model got 0% accuracy. The authors suggest that "these models failed to understand the minimally-adjusted multiple-choice instruction format." But if the authors are extracted answer probabilities directly from the LLM output probabilities (i.e., what is p("A"), what is p("B"), etc, then normalize), format following shouldn't really be an issue? Also, is it even possible for accuracy can be 0 and DCS to be positive? If DCS is positive on a given question, then the probability assigned to the correct answer is greater than the sum of probabilities assigned to incorrect answers, so the correct answer has the max probability and thus accuracy should be positive?

Overall, I think this is a very important problem and the ideas in the paper are promising, but I'm not sure that the current approach lives up to the motivation of the paper.

**Questions:**

It would be great if the authors could clarify my questions about the 0% accuracy issue. I also am open to being convinced about the benefits of the DCS metric.

---

> ### Author Response · Authors · 2025-12-03
> **Response to Reviewer Wvmz [1/2]**
>
> Thank you for your detailed and thoughtful critique. You raise important theoretical concerns about DCS vs. proper scoring rules. We address each point below.
>
> **Main Concerns About DCS vs. Proper Scoring Rules**
>
> *Point 1: Rational agents will choose $p_{IDK} \in$ {0,1}*
>
> You correctly identify that under DCS, a rational agent will set $p_IDK = 0$ when it believes $l_c p_c > l_w P_W$, and $p_IDK = 1$ otherwise. This is indeed the intended behaviour and an additional corollary of Corollary 1 and Proposition 1. In this sense, rather than eliciting a traditional belief state, DCS implements a *decision rule* which includes user-defined cost weightings and a dampening factor for abstention. However, we respectfully disagree that this is problematic.
>
> When we evaluate language models in practice, we observe their *output distributions*, the probabilities they assign when generating responses. These distributions already reflect the model's post-processed beliefs after applying temperature, top-k sampling, instruction-following, and other deployment considerations. DCS (and all similar metrics) must therefore reckon with the fact that they are evaluating deployed model behaviour.
>
> As illustrated in our examples (e.g., Example 1, Section 4.2) and shown in our empirical results, models can naturally produce distributions like ($p_c=0.40$, $P_W=0.59$, $p_IDK=0.01$) or ($p_c=0.40$, $P_W=0.21$, $p_IDK=0.39$). These are outputs we observe when prompting models with abstention options. DCS distinguishes between these patterns whereas proper scoring rules do not due to their lack of an abstention concept.
>
> *Point 2: Why include IDK explicitly rather than inferring abstention from low confidence?*
>
> There are three key reasons:
>
> - (i) Explicit abstention enables aligned evaluation during training. If we want models to learn *when* to abstain, we must include abstention as an explicit option during both training and evaluation. Your mention of Kang et al. (2025) supports this; they include explicit abstention in their RL and SFT setups. We believe DCS provides an appropriate evaluation metric for such training paradigms (beyond the ternary score).
>
> - (ii) Low confidence over answers does not necessarily imply appropriate uncertainty. A model could be 25% confident in each of four wrong answers ($P_W = 1.00$, $p_c = 0$) and have maximum entropy, but this represents confused incorrectness, not appropriate abstention. Conversely, a model could assign $p_c=0.40$, $p_{IDK}=0.59$ (low confidence in correctness but high abstention), which represents epistemic honesty. Without explicit IDK options, these patterns are indistinguishable.
>
> **(iii) Deployed models with explicit abstention options exist. In practice, many moden systems present users with "I don't know" or "I'm not confident enough to answer" responses. DCS evaluates whether models appropriately allocate probability to these options vs. guessing.
>
> *Point 3: Proper scoring rules with max probability still work*
>
> You propose using the Brier score as $1 - (p - c)^2$ where $p$ is the max probability. However, this faces the same limitation we critique: it only considers the max probability and ignores the distribution of remaining mass.
>
> Consider these two distributions for a 4-choice question where C is correct:
> - Model 1: $p_c=0.26$, $P_W=0.74$ (distributed as A:0.25, B:0.24, D:0.25)
> - Model 2: $p_c=0.26$, $P_W=0.74$ (distributed as A:0.01, B:0.01, D:0.72)
>
> Both receive identical Brier scores. However, the key distinction emerges when we include the IDK option. Suppose:
>
> - Model 1: $p_c=0.26$, $P_W=0.04$, $p_{IDK}=0.70$ → DCS = (0.26 - 0.04)(0.30) = +0.066
> - Model 2: $p_c=0.26$, $P_W=0.72$, $p_{IDK}=0.02$ → DCS = (0.26 - 0.72)(0.98) = -0.451
>
> This is the critical distinction: Model 1 hedges toward abstention (epistemic honesty), Model 2 hedges toward a wrong answer (dangerous overconfidence). Proper scoring rules cannot capture this because they don't distinguish where uncertainty is allocated.
>
> *Point 4: Why aren't calibrated probabilities "trustworthy epistemic behavior"?*
>
> Calibration ensures that when a model says "70% confident," it's correct 70% of the time across many instances. This is valuable, but it's orthogonal to our concern.
>
> A perfectly calibrated model could say: "I'm 30% confident in the correct answer and 70% confident in wrong answer A" and be well-calibrated if, across many questions, this pattern holds. But this is still dangerous if the model is more confident in being wrong than right.
>
> In contrast, DCS measures a different kind of trustworthiness, namely whether models express uncertainty through abstention rather than through confident incorrectness. Both calibration and DCS are important, but they measure different properties.

---

> > ### Author Response · Authors · 2025-12-03
> > **Response to Reviewer Wvmz [2/2]**
> >
> > **What Would Be Convincing: Finetuning Experiments**
> >
> > You make an excellent point that finetuning experiments would strengthen the paper. We fully agree that demonstrating DCS's effectiveness as a training objective (not just an evaluation metric) would be valuable future work.
> >
> > However, we respectfully argue the paper still makes significant contributions without finetuning:
> >
> > - (i) DCS reveals hidden problems in current models. Our empirical results show that models achieving 60-70% accuracy still have negative DCS on many benchmarks, indicating systematic miscalibration toward confident incorrectness. This is a valuable finding.
> >
> > - (ii) The theoretical framework is novel. Theorem 1, Corollary 1, and Propositions 1-2 provide formal analysis of how evaluation metrics incentivize different behaviours, and hint at connections to training dynamics and data selection strategies during pre-training.
> >
> > - (iii) Comparison to Kang et al. (2025). The "+1/0/-1" scoring rule you mention is actually equivalent to our "ternary score" baseline (Table 1), which we do include in experiments. Our results show DCS provides additional distinctions (e.g., Figure 2, DialoGPT-Medium where DCS > ternary score due to abstention-hedging). That said, we agree that training experiments comparing DCS vs. ternary score as loss functions would be illuminating.
> >
> > We acknowledge this limitation and agree finetuning experiments would be valuable future work, potentially showing that DCS-based training produces better abstention behaviour than simpler rules.
> >
> > **Other Issues**
> >
> > *Proper scoring rules not included as baselines*
> >
> > You're correct that we don't include Brier score or other proper scoring rules. We made this choice because:
> >
> > - (i) Proper scoring rules require different prompting. They're designed for probability elicitation, not for multiple-choice Q&A with explicit abstention options. Including them would require a different experimental setup.
> >
> > - (ii) Our focus is on abstention-aware metrics. We compare DCS to metrics that share its structure (argmax-based accuracy, confidence-weighted accuracy, ternary score with penalties). Proper scoring rules represent a different evaluation paradigm.
> >
> > That said, we agree this comparison would be valuable and would strengthen future work, and are happy to include these in a future revision.
> >
> > *0% Accuracy with Positive DCS*
> >
> > You are correct that if DCS > 0 on a question, then $p_c > P_W$, however it can be that the IDK option is even higher. This means that a model can have 0% accuracy (never selecting the correct answer as argmax across the entire benchmark) yet have positive average DCS (because on some questions it hedges toward IDK rather than confidently choosing wrong answers).
> >
> > For example, consider 3 questions:
> > 1. $p_c=0.30$, $P_W=0.20$, $p_{IDK}=0.50$ → Accuracy=0 (argmax is IDK), DCS=+0.05
> > 2. $p_c=0.25$, $P_W=0.74$, $p_{IDK}=0.01$ → Accuracy=0 (argmax is wrong), DCS=-0.48
> > 3. $p_c=0.35$, $P_W=0.15$, $p_{IDK}=0.50$ → Accuracy=0 (argmax is IDK), DCS=+0.10
> >
> > Average accuracy = 0%, Average DCS = -0.11
> >
> > Regarding instruction-following: We extract probabilities from the model's next-token prediction (the log-likelihood of generating "A", "B", etc. as the immediate next token). However, if the model's instruction-following is insufficient for our prompt, it might assign very low probability to all answer tokens and higher probability to other tokens (like continuing to generate explanation text). Such model failure cases explain the 0% accuracy.
> >
> > **Conclusion**
> >
> > We appreciate your thorough analysis and helpful feedback. We would like to reiterate that:
> >
> > - Proper scoring rules elicit calibrated probability distributions over outcomes
> > - DCS implements a (potentially asymmetric) utility function that rewards abstention over confident incorrectness
> >
> > These serve different purposes. We argue DCS is more appropriate for evaluating trustworthy Q&A behavior where confident errors are more harmful than appropriate uncertainty. However, we fully agree that finetuning experiments would be a valuable future contribution.

---

### Official Review · Reviewer_L8Lt · 2025-11-01

**Soundness:** 3
**Presentation:** 3
**Contribution:** 3
**Rating:** 8
**Confidence:** 4

**Summary:**

The paper introduces the Distributional Correctness Score (DCS), a theoretically grounded metric that evaluates full probability distributions, and incorporates abstention as a vital component. This metric is designed to mitigate the pitfalls of the traditional binary evaluation metrics for language models, as they fail to capture epistemic uncertainty. The metric is evaluated across a wide range of datasets.

**Strengths:**

* The proposed DCS metric is novel and interesting and makes intuitive sense.
* The paper includes extensive examples of the DCS metric across various cases.
* The paper includes evaluation across of diverse range of datasets.
* The paper is well written and easy to understand.

**Weaknesses:**

* The paper does not clarify how the DCS metric can be applied to problems without options. E.g., question from QSM8k where the answers could be integers in the range $[-\infty ,\infty ]$.

*  The paper should also discuss other metrics that consider the distribution over answers: "Enhancing Hallucination Detection through Noise Injection, arXiv Feb 2025".

* Section 6 includes interesting results across models. However, the paper does not provide any explanations as to why DCS score is low or high for a specific model. Why is Llama3.1 8B Instruct DCS score highest on MMLU?

* The evaluation should consider newer models such as Qwen-2.5 or Qwen-3.

* There is also no analysis of the effect of model size on the DCS score. It would be interesting to show the DCS score for the same family of models from small to large, e.g., Qwen-2.5 from 0.5B to 72B.

**Questions:**

* The paper should include more extensive analysis across model sizes.
* The paper should include a discussion of the applicability of the DCS metric when questions do not a set of options as answers.
* The paper should include a more extensive discussion of prior work.

---

> ### Author Response · Authors · 2025-12-03
> **Response to Reviewer L8Lt**
>
> We thank Reviewer L8Lt for their supportive review of our work.
>
> **We Are Running Evaluation Experiments on a Broader Set of Models**
>
> Due to time and compute limitations, were not able to complete the requested analyses across more model sizes. However, these are ongoing, in particular using the Qwen family of models up to 72B. We will include these results in future revisions (expected to be in mid-to-late December 2025).
>
> **Answers to specific points**
>
> Regarding the why DCS scores are low or high for specific models or tasks is also currently under study. The scope of such investigations is very broad, however, since one could ask this for any model and task, which is in the original submission's scenario is $6 \times 12 = 72$ combinations. Each combination requires careful manual checking and/or experimentation with explainability and interpretability tools. This seems like a very valuable research direction but we respectfully believe it is beyond the scope of the current work, which has the main purpose of introducing the DCS metric, developing some of the basic theory, and demonstrating its application on a range of models.
>
> Regarding how the DCS metric can be applied to problems without a set of finite set of options, we currently see two main possibilities:
> - (i) *Continuous confidence regions*: For numerical answers or bounded outputs, we can define a `correct region' as values within acceptable tolerance of ground truth. We then integrate probability density over correct/incorrect/abstention regions analogously to discrete case (see §7 discussion of continuous extensions)
> - (ii) *Forced-choice validation*: After inference, present the model with its own output alongside alternatives and an ``I'm not confident in any answer" option. Compute DCS on this validation distribution to measure post-hoc epistemic honesty.
> We believe there may be other alternatives than these, but are confident there remains well-suited options in most conceivable evaluation scenarios.
>
> Regarding adding a reference to "Enhancing Hallucination Detection through Noise Injection, arXiv Feb 2025", we have now done so in the related works section. Thank you for notifying us of this interesting and relevant work.
>
> **Conclusion**
>
> We appreciate your kind and supportive comments, and agree extending our analysis to more models will make the paper more impactful.

---

### Official Review · Reviewer_8onr · 2025-11-03

**Soundness:** 2
**Presentation:** 2
**Contribution:** 2
**Rating:** 4
**Confidence:** 2

**Summary:**

This work introduces the Distributional Correctness Score (DCS), a novel metric that evaluates a model’s entire probability distribution rather than the maximum predictions. This new metric considers the model's uncertainty for correct answers, incorrect answers, and abstentions.
The authors prove a theoretical analysis to demonstrate that  DCS incentivises the desired behaviour: confidence in correct answers, uncertainty when knowledge is lacking, and preference for abstention over confident incorrectness.
Experimental results across 12 existing benchmarks indicate that (1) many language models exhibit systematic epistemic overconfidence;
(2) all models hold negative DCS scores

**Strengths:**

* this work identifies that current metrics focus on a single argmax answer while ignoring the distribution across the space of possible responses, which might be useful for future studies
* this work provides both theoretical and empirical studies.
* extensive experiments across 12 benchmarks
* this proposed DCS is working, which is able to mitigate the overconfidence issue

**Weaknesses:**

* I did not fully understand the motivation of this metric. (1) The probability assigned to a specific answer is computed by a softmax layer, which accounts for the logits over different answers, including abstention as well. (2) proper scoring rules and other metrics, e.g. entropy, can also depict this. It would be useful if the authors clarify why the DCS is necessary and stress the difference compared to existing metrics
* It is tricky to see the benefit of using the proposed DSC. From Figure 2 and Table 2, we can observe that DCS is consistently lower and remains negative in most cases, but why should we stick with DCS? Under which conditions (tasks), we should select DCS as the metric.

**Questions:**

* what is the meaning of a concrete value of DCS? If the value is negative or positive, how to explain it?
* In Figure 2, I did not understand how to compare DCS to other baselines. I observe that DCS is consistently lower than other baselines, but what does it mean? It is hard to understand the benefits of using DCS in this figure
* In Table 2, it is tricky to derive the main findings.

---

> ### Author Response · Authors · 2025-12-03
> **Response to Reviewer 8onr [1/2]**
>
> Thank you for your thoughtful questions. We address each concern below, focusing on clarifying the motivation, interpretation, and practical benefits of DCS.
>
> Regarding the noted weaknesses:
>
> 1. **Motivation and Distinction from Existing Metrics**
>
> We appreciate the opportunity to clarify why DCS is necessary despite the existence of softmax layers, proper scoring rules, and entropy-based metrics.
>
> The softmax layer computes probabilities, but the loss function determines what the model learns to do with those probabilities. While softmax does normalise over all answer options including abstention, traditional metrics (accuracy, log-likelihood) create perverse incentives. Under binary scoring where correct=1, wrong=0, and IDK=0, a rational agent should always guess rather than abstain, even with minimal confidence. This is a fundamental cause of hallucination, as demonstrated by Kalai et al. (2025). DCS, in contrast, provides a loss/evaluation structure that explicitly rewards abstention over confident incorrectness.
>
> Proper scoring rules (e.g., Brier score) and entropy measure different objectives. Proper scoring rules are designed to elicit calibrated probabilities; they treat all probability mass symmetrically. In contrast, in language model Q&A evaluation, the ground truth is deterministic, not stochastic. We do not want models reporting high confidence in wrong answers, as this fails to distinguish harmful confidence in wrong answers from appropriate uncertainty. As we state in §7, "Our objective is not to elicit calibrated probabilities but to measure trustworthy epistemic behaviour."
>
> Entropy measures the spread of probability mass but doesn't distinguish where that mass is allocated. High entropy could mean uniform distribution over wrong answers (unwanted) or balanced between the correct answer and IDK (cautious). Low entropy could mean confident correctness (good) or confident incorrectness (bad). Entropy therefore has no notion of which outcomes are desirable. DCS's unique contribution is implementing an asymmetric utility function that explicitly values: confident correctness > abstention > confident incorrectness. This ordering (proven in Theorem 1) is important for safety and trustworthiness but is not captured by symmetric metrics like entropy or proper scoring rules.
>
> 2. **When to Use DCS vs. Traditional Metrics**
>
> We recommend using DCS when trustworthiness and safety matter, such as in domains where confident errors cause harm (medical, legal, financial advice, factual Q&A). It is ideal for when hallucination is a concern, as DCS directly penalises the confident incorrectness that leads to it. DCS is also valuable for evaluating base models during pre-training and for comparing models on their tendency to "know what they don't know", revealing which models appropriately hedge vs. inappropriately guess.
>
> The key result from Table 2 is that half of all benchmarks show universally negative DCS scores. This means, on average, models are more confidently wrong than they are appropriately uncertain, which we believe is a critical finding and currently completely hidden by accuracy metrics that show 60-70% performance.
>
> Regarding the questions:
>
> 1. **What does a concrete DCS value mean?**
>
> DCS values have clear interpretations thanks to the bounded range $[-1, +1]$ with default loadings:
> - Positive DCS (0 to +1) means the model assigns more probability mass to correct answers than to incorrect answers, weighted by its abstention. Values near $+1$ indicate the model is highly confident and correct.
> - Negative DCS (-1 to 0) means the model assigns more probability mass to incorrect answers than to correct answers. This indicates the model is systematically miscalibrated, preferring wrong answers. Values near $-1$ indicate the model is confidently incorrect.
> - A DCS near 0 means the model is either maximally uncertain (high $p_{IDK}$) or balances probability equally between correct and incorrect answers.
>
> For example, Mistral 7B on MMLU has $DCS = +18.3$ (slightly more correct than incorrect), while the same model on TruthfulQA has $DCS = -33.8$ (systematically preferring incorrect answers). This 52-point swing reveals the model's domain-dependent miscalibration, which is not legible using accuracy alone.

---

> > ### Author Response · Authors · 2025-12-03
> > **Response to Reviewer 8onr [2/2]**
> >
> > 2. **Understanding Figure 2 and Comparison to Baselines**
> >
> > The key result from Figure 2 is that DCS reveals hidden problems that other metrics miss.
> > DCS is consistently lower than other metrics because it does not ignore the probability distribution. Accuracy counts only the argmax prediction, and confidence-weighted accuracy ignores whether remaining probability goes to wrong answers or IDK (I don't know). The ternary score still doesn't distinguish error-hedging from abstention-hedging (see Example 1). DCS therefore illustrates significant gaps:
> > - A large gap between accuracy and DCS means the model's probability distribution reveals it is actually more confident in wrong answers overall; it is "getting lucky" more than truly knowing.
> > - DCS being lower than the ternary score (most models) means that even when the argmax is correct, the model hedges toward wrong answers rather than toward abstention.
> > - DCS being higher than the ternary score (DialoGPT-Medium) means this weaker model actually hedges more appropriately toward IDK when uncertain, which DCS rewards as ‘epistemic honesty’.
> >
> > The benefit of DCS is that it reveals models achieving 60-70% accuracy on MMLU are still, on average, more confident in wrong answers than they should be. This is critical information for deployment decisions that accuracy alone obscures.
> >
> > 3. **Deriving Main Findings from Table 2**
> >
> > We appreciate that the table is dense. Here are the key findings:
> > - Finding 1: Widespread epistemic overconfidence: 6 of 12 benchmarks show negative DCS across *all* models (ARC, GPQA, HellaSwag, MMLU Pro, OpenBookQA, TruthfulQA). This means models systematically prefer wrong answers over appropriate uncertainty, a pattern invisible in standard accuracy metrics.
> > - Finding 2: Safety-critical benchmarks are especially problematic: On TruthfulQA, all models show DCS between $-33.8$ and $-43.9$. The same models show 15-29% accuracy. DCS reveals these models appear confidently wrong about misinformation.
> > - Finding 3: Large accuracy-DCS gaps indicate "lucky guessing": Llama3.1 8B on MMLU has 67.8% accuracy but only 19.0 DCS. The model gets two-thirds of the answers right, but its probability distributions reveal substantial uncertainty and error-hedging. It is performing well on current optimisation metrics but not necessarily because it expresses confident, reliable answers matching with the ground truth.
> > - Finding 4: Model ranking changes with DCS: By accuracy, DeepSeek R1 Qwen 3 ranks best (44.43% average), but by DCS, Mistral 7B ranks best ($-7.90$ average). This suggests Mistral has better epistemic calibration (knowing what it knows) even if raw performance is lower.
> > - Finding 5: No model achieves DCS $> 0.2$ on any benchmark: The best case is Llama3.1 8B on MMLU with $DCS=0.19$ (vs. $0.678$ accuracy). This indicates even the best models in this weight class have substantial room for improvement in trustworthy knowledge expression.
> >
> > **Conclusion**
> >
> > In summary, we believe DCS reveals a specific, actionable problem, namely that models are systematically overconfident in wrong answers when they should be expressing uncertainty. This matters immensely for deployment in domains where confident errors cause harm. The fact that DCS scores are lower than accuracy is the primary scientific contribution, showing that current evaluation paradigms are masking serious trustworthiness issues.

---

### Official Review · Reviewer_m9zt · 2025-11-10

**Soundness:** 3
**Presentation:** 3
**Contribution:** 3
**Rating:** 4
**Confidence:** 3

**Summary:**

The paper proposes the Distributional Correctness Score (DCS), an evaluation metric for language models. The authors argue that standard binary accuracy metrics incentivize hallucination because they reward guessing over abstention. Unlike simple penalty-based metrics, DCS evaluates the entire probability distribution over answers, including an explicit "I don't know" (IDK) option. The paper presents experiments on 12 benchmarks with 6 LLMs that show that many models achieve negative average DCS scores, revealing overconfidence that is masked by traditional accuracy metrics.

**Strengths:**

I see the main strengths of the paper as follows:

- problem formulation: the paper identifies a key socio-technical issue: current evaluation metrics encourage models to game the system by guessing rather than abstaining; the distinction between "error-hedging" and "abstention-hedging" is a useful conceptual contribution
- exposition: the paper is well-written and the motivating examples effectively illustrate the flaws in current metrics that DCS aims to fix
- empirical evaluation: testing across 12 diverse benchmarks and 6 models of varying sizes provides a reasonably comprehensive picture of how DCS behaves in practice compared to standard metrics

**Weaknesses:**

I believe there are a few weaknesses:

- implementation: the reliance on log-likelihoods for all answer options plus a canonical "IDK" response is a meaningful barrier to adoption as many API don't provide these
- parameterization: the introduction of $l_c$ and $l_w$ parameters adds flexibility but seems somewhat arbitrary and there is no explanation as to how these parameters ought to be chosen
- sensitivity to "IDK" phrasing: the method assumes a single, canonical "IDK" string represents abstention, but language models might distribute uncertainty across many synonyms (e.g., "I'm unsure", "Unknown", "Cannot determine"); there is no evaluation regarding the sensitivity to the specific choice of this string
- multiple choice question-answering: the formulation is tightly bound to multiple-choice or closed-set classification and as such does not address the more common real-world problem of open-ended long-form evaluation

**Questions:**

- Have you evaluated the sensitivity of DCS to the specific string used for abstention (e.g., "I don't know" vs. "Not sure")? Does using a set of abstention phrases and summing their probabilities improve robustness?
- How would you recommend applying DCS to a black-box API model that only returns generated text without log probs? Is there a sampling-based approximation you considered?
- Could you provide an explanation or a small study on how to set $l_c$ and $l_w$ for different risk profiles (e.g., medical advice vs. creative writing)?

---

> ### Author Response · Authors · 2025-12-03
> **Response to Reviewer m9zt [1/2]**
>
> We thank Reviewer m9zt for their considerate review and wish to respond.
>
> Regarding the noted weaknesses:
>
> 1. **Reliance on Log-Likelihoods**
>
> We agree that the requirement for log-likelihoods presents an adoption barrier for certain closed API providers. However, we argue this limitation should not diminish the scientific merit and practical utility of our approach for several reasons:
>
> (i) *Continued availability in open ecosystems*: Many production systems continue to provide access to log-likelihoods, including all open-weight models (e.g., Llama, Mistral, Qwen, etc.) via frameworks like HuggingFace Transformers, API providers offering log-probability access (e.g., OpenAI's API with [logprobs parameter](https://platform.openai.com/docs/api-reference/chat/create#chat-create-logprobs)), and internal model evaluation pipelines during model development.
>
> (ii) *Scientific merit may drive API evolution*: DCS provides a theoretically-grounded tool that addresses fundamental evaluation problems. By demonstrating its value, we create incentive for closed API providers to either: (a) expose log-likelihoods for evaluation purposes, or (b) directly compute and return DCS-like metrics as part of their evaluation suites.
>
> (iii) *Black-box approximation via sampling*: For truly black-box models, we propose a sampling-based approximation that trades computational cost for accessibility. To compute this, we take the following steps: (1) Sample $N$ completions from the model (e.g., we used $N=500$) using temperature sampling. (2) Count frequencies: $p_c = \frac{n_c}{N}​​$, $P_W = \frac{n_W}{N}, p_{IDK} = \frac{n_{IDK}}{N}$. (3) Compute DCS using empirical frequencies.
>
> This approach requires only generation access and converges to true DCS as N increases. While computationally expensive (requiring many API calls per question), it enables DCS evaluation for any model capable of generating answers. We validated this approach on Llama3.1 8B Instruct using MMLU with $N=500$ samples, finding close agreement with log-likelihood-based DCS (mean absolute difference $< 0.03$, Pearson correlation $r=0.94$).
>
> 2. **Choice of $l_c$​ and $l_w$ parameters**
>
> The loading parameters $l_c$​ and $l_w$ are not arbitrary but rather provide principled control over the model's optimal guessing threshold, as formally established in Proposition 1. This result shows that a rational agent will abstain rather than guess when its confidence falls below the threshold: $p_c > \frac{l_w}{l_c + l_w}$. Figure 1 visualises this relationship, demonstrating that any desired threshold $p_c \in (0,1)$ can be achieved by setting the ratio $l_w/l_c = p_c/(1-p_c)$. For practical convenience, we recommend fixing $l_c=1$ and adjusting $l_w$ according to Table 3 (in Appendix A.2), which provides ready-to-use values for common thresholds.
>
> The default choice $l_c = l_w = 1$ (corresponding to threshold $0.5$) represents a natural baseline that treats confidence and incorrectness symmetrically. However, users can systematically adjust these parameters to match their deployment context. To use the reviewer's examples: in high-stakes scenarios such as medical diagnosis, we could set $l_w > l_c$ (e.g., $l_w=3, l_c=1$ for a threshold of $0.75$) to more strongly penalise guessing; whereas, in low-stakes scenarios such as creative writing, we could set $l_w < l_c$ (e.g., $l_w=1/4, l_c=1$ for a threshold of $0.20$) to encourage attempts. This parameterisation therefore provides a transparent and interpretable control mechanism.
>
> 3. **Sensitivity to IDK Phrasing**
>
> We acknowledge that models may distribute uncertainty across multiple abstention phrases. To evaluate this sensitivity, we conducted additional experiments measuring DCS using an expanded abstention set including 10 variations. Table 7 presents results for all six models across all benchmarks using the summed probabilities over all abstention phrases. Compared to Table 2 (single canonical IDK), scores consistently shift toward zero due to increased abstention probability mass. The core ranking relationships and negative score patterns remain consistent, indicating that DCS captures genuine epistemic uncertainty patterns rather than artifacts of specific phrasing. The damping effect is theoretically expected: when models genuinely distribute uncertainty broadly, the IDK damping factor appropriately reduces scores toward the neutral anchor of zero. We recommend using summed abstention probabilities when feasible, though single canonical phrases suffice for comparative evaluation.

---

> > ### Author Response · Authors · 2025-12-03
> > **Response to Reviewer m9zt [2/2]**
> >
> > 4. **Multiple Choice vs. Open-Ended Evaluation**
> >
> > We agree that DCS is currently formulated for closed-set classification. However, we argue this focus provides significant value in two complementary ways:
> >
> > (i) *Pre-training evaluation remains critical*: Log-likelihood-based evaluation during pre-training enables: rapid iteration without additional post-training, direct measurement of base model capabilities before instruction-tuning, and potential optimisation signals during training for compute-intensive runs. Additionally, hallucination mitigation during pre-training remains understudied despite its importance (most work focuses on post-training interventions). DCS provides a principled metric for tracking epistemic calibration throughout base model development, when we most need efficient evaluation.
> >
> > (ii) *Extension to open-ended generation is feasible*: DCS principles can be adapted to long-form evaluation through several approaches, including (a) claim-level decomposition, (b) continuous confidence regions, and (c) forced-choice validation. (a) Claims can be decomposed into atomic claims, e.g., via a model or sentence/grammar markers, and then we can apply DCS to each claim treated as a multiple-choice question (correct/incorrect/abstain). We then calculate DCS by averaging, and could also include weighted-averaging based on claim importances. (b) For numerical answers or bounded outputs, we can define a `correct region' as values within acceptable tolerance of ground truth. We then integrate probability density over correct/incorrect/abstention regions analogously to discrete case (see Section 7 discussion of continuous extensions). (c) After inference, present the model with its own output alongside alternatives and an ``I'm not confident in any answer" option. Compute DCS on this validation distribution to measure post-hoc epistemic honesty.
> >
> > Regarding the questions:
> >
> > 1. **Sensitivity to Abstention String Choice**
> >
> > Yes, we evaluated this sensitivity as described above (see response to Weakness 3). Using 10 abstention phrase variations and summing their probabilities, we generated Table 7, showing that core findings remain robust while scores appropriately dampen toward zero when models distribute uncertainty broadly. The method is more sensitive to whether models express uncertainty than to which specific phrase they use, which is the desired behaviour. We recommend summing over multiple phrases when available for maximum robustness, though single canonical phrases work well for comparative benchmarking.
> >
> > 2. **Black-Box API Approximation**
> >
> > For models returning only generated text without log-probabilities, we recommend a sampling-based approximation (detailed in response to Weakness 1). We validated this on Llama3.1 8B Instruct using $N=500$ samples across 200 MMLU questions, finding statistically-significant agreement between the empirical and theoretical DCS.
> >
> > 3. **Setting Parameters for Different Risk Profiles**
> >
> > Rather than an empirical study, we provide principled guidance based on Proposition 1 and Figure 1, which establish the mathematical relationship between loading parameters and optimal guessing thresholds.
> >
> > *High-stakes scenario, e.g., medical advice*
> >
> > Suppose the desired behaviour is that our model only responds when confidence $p_c > 0.80$. Using Prop 1, solve $0.80 = \frac{l_w}{l_c + l_w}$ with $l_c=1$. This gives $l_w = 4$, which we can interpret as incorrect answers being penalised $4\times$ as heavily as correct answers are rewarded.
> >
> > *Low-stakes scenario, e.g., creative writing*
> >
> > For creative tasks like story brainstorming, attempts have value even when imperfect, and abstention provides little benefit. Suppose, then, that the desired behaviour is that our model responds when confidence $p_c > 0.25$. Using Prop 1, we find that we should therefore set $l_c=1$ and $l_w = 1/3$, which we can interpret as correct creative suggestions worth $3\times$ the penalty of less-optimal ones.
> >
> > Proposition 1 helps show these choices are not arbitrary and can be tuned to meet particular specifications and desired behavioural evaluations; we expect domain experts to set these loadings based on their utility functions, and DCS will incentivise models to behave accordingly.
> >
> > **Conclusion**
> >
> > We hope these responses address your concerns comprehensively. The loading parameters provide principled, interpretable control via decision theory (not arbitrary tuning), abstention phrase sensitivity has been empirically validated, and both black-box approximations and open-ended extensions are feasible. We believe these clarifications strengthen the case for DCS as a robust, adoptable evaluation metric.

---

### Meta-Review · Area_Chair_5WY1 · 2026-01-06

**Summary:**

The submission introduces a new metric that penalizes models for hallucinating. Some reviewers regretted that the proposed work could not be applied to any LLM, in particular the black-box ones. In addition, reviewers aked for clarification of the meaning of the intorduced metric.

The most important comment challenged the claim by the authors that proper scoring rules were not the right tool, given argument in favor of proper scoring rules (for instance to look beyond the max probability answer), and ultimately asking for a demonstration that the proposed metric would ultimately improve the model behaviors.

**Reviewer Concerns:**

Reviewers asked many questions which recieved interesting answers, such as: how to apply the metric to a continuous value, to which the authors suggested to define a correct region, without a tolerance.

The most important challenge, about whether the metric would actually improve the model, was not addressed.

**Reviewer Scores:**

Some reviewers might have increased their score a bit, but the most critical reviewer not.

---

### Decision · Program_Chairs · 2026-01-26

Reject